# Seasonal changes in proportion of cardiac surgeries associated with diabetes, smoking and elderly age

Ferenc Peták[1]*, Barbara N. Kovács[1,2⊙], Szilvia Agócs[2,3], Katalin Virág[1], Tibor Nyári[1], Andrea Molnár[1,2], Roberta Südy[1,2], Csaba Lengyel[4], Barna Babik[2,3]

1 Department of Medical Physics and Informatics, University of Szeged, Szeged, Hungary, 2 Department of Anesthesiology and Intensive Therapy, University of Szeged, Szeged, Hungary, 3 Department of Internal Medicine and Cardiology Center, Cardiac Surgery Unit, University of Szeged, Szeged, Hungary, 4 Department of Internal Medicine, University of Szeged, Szeged, Hungary

⊙ These authors contributed equally to this work.
* petak.ferenc@med.u-szeged.hu

**Data Availability Statement:** All relevant data are within the paper and its Supporting Information files.

## Abstract

### Background

Seasonal variations in the ambient temperature may affect the exacerbation of cardiovascular diseases. Our primary objective was to evaluate the seasonality of the monthly proportion of cardiac surgeries associated with diabetes, smoking and/or elderly age at a tertiary-care university hospital in East-Central Europe with a temperate climate zone. As a secondary objective, we also assessed whether additional factors affecting small blood vessels (smoking, aging, obesity) modulate the seasonal variability of diabetes.

### Methods

Medical records were analyzed for 9838 consecutive adult patients who underwent cardiac surgery in 2007–2018. Individual seasonal variations of diabetes, smoking, and elderly patients were analyzed monthly, along with the potential risk factors for cardiovascular complication. We also characterized whether pairwise coexistence of diabetes, smoking, and elderly age augments or blunts the seasonal variations.

### Results

Seasonal variations in the monthly proportion of cardiac surgeries associated with diabetes, smoking and/or elderly age were observed. The proportion of cardiac surgeries of non-elderly and smoking patients with diabetes peaked in winter (amplitude of change as *[peak-nadir]/nadir*: 19.2%, p<0.02), which was associated with increases in systolic (6.1%, p<0.001) and diastolic blood pressures (4.4%, p<0.05) and serum triglyceride levels (27.1%, p<0.005). However, heart surgery in elderly patients without diabetes and smoking was most frequently required in summer (52.1%, p<0.001). Concomitant occurrence of diabetes and smoking had an additive effect on the requirement for cardiac surgery (107%, p<0.001), while the simultaneous presence of older age and diabetes or smoking eliminated seasonal variations.

**Funding:** Hungarian Scientific Research Fund (OTKA-NKFIH K13803). The funders had no role in study design, data collection and analysis, decision to publish, or preparation of the manuscript.

**Competing interests:** The authors have declared that no competing interests exist.

**Abbreviations:** BMI, body mass index; HbA1c, hemoglobin A1c; SM, smoking; T2DM, type 2 diabetes mellitus.

## Conclusions

Scheduling regular cardiovascular control in accordance with periodicities in diabetes, elderly, and smoking patients more than once a year may improve patient health and social consequences.

## Trial registration

NCT03967639.

## Background

Diabetes mellitus is a complex metabolic disease that can affect vital functions [1]. Cardiovascular complications, such as atherosclerotic cardiovascular and heart failure, are the leading cause of morbidity and mortality in patients with type 2 diabetes mellitus (T2DM), which is the most common form of the disease [2, 3]. Treatment of cardiovascular comorbidities comprises drug therapy; however, progression and/or exacerbation of these diabetes-related circulatory complications often require surgical intervention.

It is well established that the incidence of severe cardiovascular diseases exhibits a seasonal pattern, with more frequent relative occurrence during winter [4–9]. Several factors may contribute to these seasonal variations, such as activation of the sympathetic nervous system and increased cathecolamines [10], elevated serum cholesterol levels [11], prothrombotic shift in the hemostatic system via elevated fibrinogen levels [12, 13], decreased physical activity [4], and vitamin D deficiency [14]. Most of these pathological processes are influenced by diabetes [2]. Consequently, seasonal augmentation of clinical signs and symptoms can be anticipated in patients with T2DM, which may require alterations in treatment strategy involving the requirement for cardiovascular surgery.

Therefore, the primary objective of the present study was to reveal whether the proportion of cardiac surgeries associated with diabetes requiring heart surgery exhibits seasonal variations, peaking during winter. To address this goal, we evaluated the monthly proportion of cardiac surgeries for patients with diabetes within a 12-year period at the cardiac surgery unit of a tertiary-care university hospital. As a secondary objective, we also evaluated whether factors affecting small blood vessels (smoking, aging, and obesity) modulate the seasonal variability of T2DM along with the potential risk factors for cardiovascular complications (blood pressure, serum triglyceride, cholesterol and glucose levels). The rationale of the study is related to the fact that worsening and/or exacerbation of cardiovascular complications necessitating surgery can often be prevented with appropriate medical treatments in patients with diabetes. Thus, exploration of this cold-related seasonal phenomenon may elucidate the need for a more frequent patient follow-up to avoid progression with appropriately timed preventive measures.

## Methods

### Study design and population

Ethical approval for this study (no. 274/2018/a) was provided by the Human Research Ethics Committee of Szeged University, Hungary (chairperson, Prof. T. Wittmann) on January 21, 2019. The study was registered at clinicaltrials.gov (NCT03967639).

Medical records were retrospectively analyzed for all 9838 consecutive adult patients who underwent surgery at our institution (Cardiac Surgery Unit, Second Department of Internal

Medicine and Cardiology Center at the University Hospitals of Szeged, Hungary) from January 1, 2007 to December 31, 2018. Patients underwent the entire spectra of cardiac surgeries. Our clinical practice avoided a waiting list; therefore, all operations were performed within five days after establishing the requirement for surgical intervention. Patient records were discarded in case of emergency reoperations as a consequence of tamponade or acute bleeding, since these events are not related to exacerbation of cardiovascular disorders. Accordingly, patients were included in the analyses only after a primary or redo open heart surgeries.

Cardiac surgery patients were assigned to the following groups, or combinations of groups, based on hospital medical records. Patients were defined as having T2DM if their medical history included a diagnosis of T2DM and/or hemoglobin A1c (HbA1c) > 6.5%, in accordance with the diagnostic criteria of the American Diabetes Association [15]. Since almost all (99.6%) patients with diabetes had T2DM, and the etiology and pathophysiological characteristics of type 1 diabetes mellitus differs from that of T2DM, only T2DM patients were included in the analyses, with an average of 8.6 years diagnosed disease period. Among T2DM patients, 25.8% were treated with insulin. T2DM patients treated with insulin or oral antidiabeticss were pooled in the final analyses. Patients were assigned to the smoking group based on the definitions of the National Center for Health Statistics [16]: current smokers (smoked 100 cigarettes in his or her lifetime and who currently smokes cigarettes); everyday smokers (smoked at least 100 cigarettes in his or her lifetime and currently smokes every day); or ex-smokers (ceased tobacco use <12 months ago). Patient were considered to be elderly if they were older than the average life expectancy age in southern Hungary published by the Hungarian Central Statistical Office during the study period: ≥72 years for males and ≥79 years for females [17]. Obesity was classed according to the definition of the World Health Organization as body mass index (BMI) ≥30 kg/m$^2$ [18]. Individual seasonal effects of these factors were analyzed on a monthly basis. To identify the coexistence of these factors with potential additive or regressive effects on seasonal changes, the combined occurrence of statistically significant factors were also examined.

Noninvasive systolic and diastolic blood pressure values were registered at admission, and serum triglyceride, cholesterol and glucose levels were measured from venous blood samples collected from the first blood samples after arrival to the hospital.

## Monthly average temperature data

Average monthly temperature data for the study period were obtained from the database of the Hungarian Meteorological Service.

## Data processing and statistical analyses

Statistically significant differences between the study groups for continuous variables were assessed by one-way analysis of variance followed by followed by Bonferroni's post-hoc tests. Pearson's Chi-squared tests were used to evaluate differences in categorical variables. Monthly proportion of surgeries associated with the different disorders were calculated as the number of new patients for cardiac surgeries with a given risk factor (independently of the other two risk factors, alone, and in combination; e.g., T2DM alone; T2DM and smoking; T2DM, smoking, and elderly) divided by the total number of cardiac surgery patients in the same month. Seasonality of the monthly aggregated proportion of surgeries associated with the observed disorders and values for blood pressure, triglyceride, cholesterol and glucose during the study period was assessed using Walter–Elwood and negative binomial regression methods [19], assuming that data followed a sinusoidal curve with a periodicity of one year. Geometric models were used to investigate seasonality by assuming that seasonal fluctuations of an event

occur on a fixed date every year and might be described using cyclic patterns over a period of time. The power of the Walter-Elwood test is 100% [20], and the percentage of change (variation) is the main effect size for the association between seasonal variation and health parameters. The deviance statistic was used to check a goodness of fit for negative binomial regression models. Similarly, Walter and Elwood also described a goodness of fit calculation for their methods [19], which was also taken into account.

Diabetes, aging, smoking, obesity, and gender were considered as possible risk factors for cardiac surgeries. Relative change (peak–mean)/mean was calculated to quantify the severity of seasonality and two compare seasonal amplitudes. Statistical analyses were performed using Stata software package (version 17, Statacorp, College Station, Texas) and p-values < 0.05 were considered statistically significant. The charts were prepared by using SigmaPlot software package (Version 13, Systat Software, Inc. Chicago, IL, USA).

## Results

The involvement of patients in the retrospective data analyses and their group allocation is demonstrated on a CONSORT flow chart (Fig 1). The total of 9881 patients were enrolled in the study period. Twenty-seven patients were excluded from the data analyses due to incomplete data set in anthropometrical data and/or blood pressure and/or blood sample analyses. Furthermore, type 1 diabetes was diagnosed in 16 patients; they were also excluded from the analyses due to their fundamentally different diabetes phenotype than in the main population including T2DM. These considerations resulted in classifications of 9838 patients.

### Anthropometric data and clinical characteristics

The anthropometric data and main clinical characteristics of the study groups according to significant factors exhibiting seasonal variations in the overall proportion of surgeries are summarized in S1 Table in S1 File. In agreement with the worldwide proportion of patients undergoing cardiac surgery who had T2DM (30%-40%) [21], 38.4% of patients were diagnosed T2DM in the present study. In accordance with the diagnostic criteria, HbA1c was significantly higher in patients with diabetes (7.75±1.17) than in those without metabolic disorders (5.69±0.4). The preponderance of males observed in the whole study population (63.1%) was also present in each subgroup, with the exception of patients with T2DM alone (51.4%) and those without examined risk factors (52.4%). T2DM was significantly associated with higher body weight (84.4 vs. 77.6 kg, p<0.001) and BMI (31.0 vs. 28.1 kg/m$^2$, p<0.001), whereas smoking was associated with lower BMI (27.1 kg/m$^2$; p<0.001). Compared with patients without risk factors, aortic diseases were more frequent in elderly patients (p<0.05), whereas the prevalence of grown-up congenital heart diseases was lower in T2DM, smoking, and elderly patients (p<0.05). The proportion of coronary disease was generally greater in T2DM and smoking patients (p<0.05).

The total number of cardiac surgery patients included in the data analyses in each month is demonstrated on S2 Table in S1 File. These data show no significant seasonality; only the holiday seasons show a decrease in the number of patients due the limited availability of human resources available at the university hospital, however the relative frequency of the various diagnoses was not different between months.

### Seasonal variabilities: Significant risk factors

There were no statistically significant seasonal variations for gender (p = 0.81) and BMI (p = 0.75); therefore, these variables were not included in further analyses. The main types of heart disease with sufficient numbers of patients available to analyze seasonal changes revealed

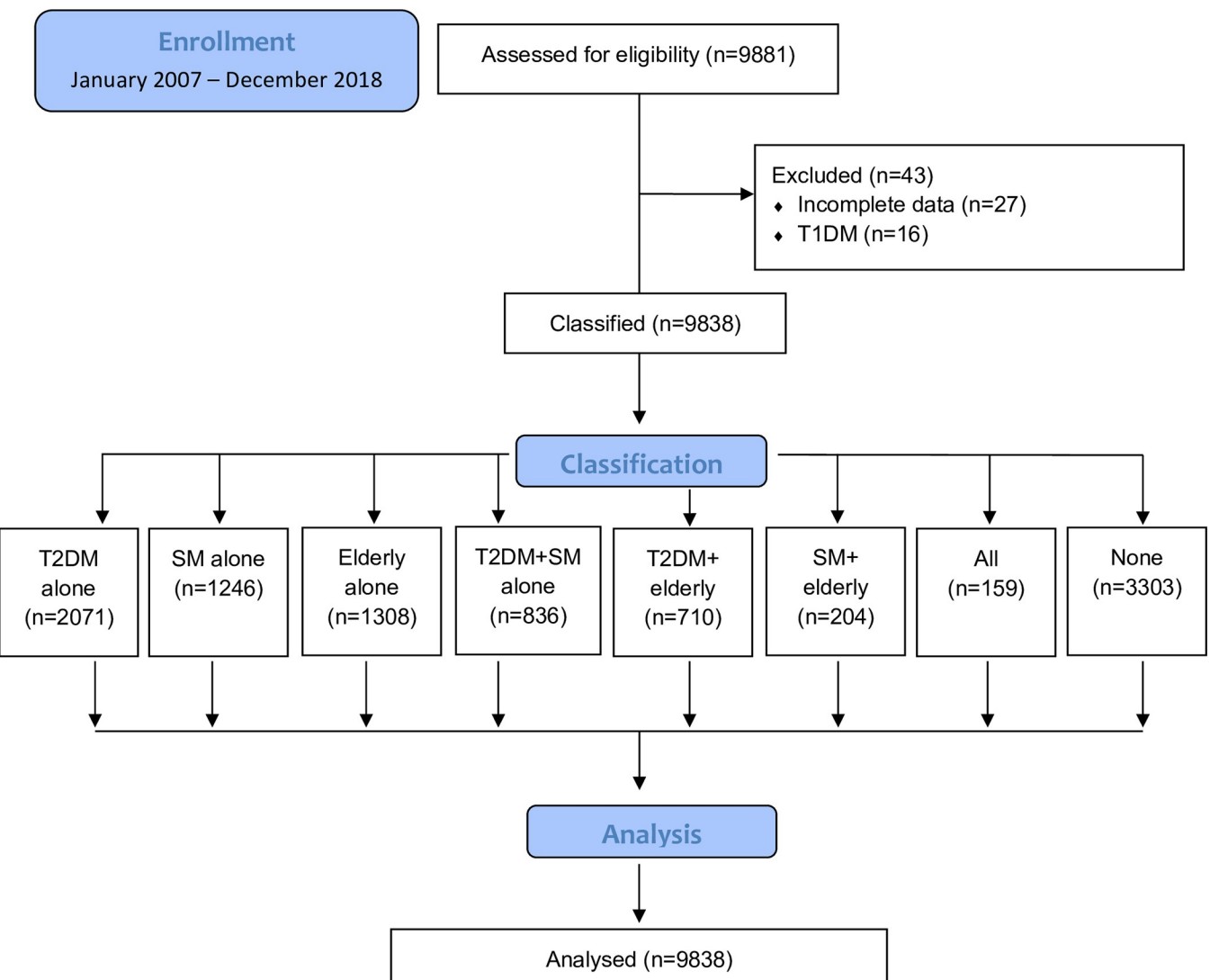

**Fig 1. Consort flowchart.** Group allocation ana analyses of cardiac surgery patients with diabetes mellitus only (T2DM alone), smoking (SM alone), and aging (Elderly alone). Groups containing pairwise (T2DM + SM, T2DM + Elderly, and SM + Elderly) and concomitant combination ("All") significant factors were also separated. "None" denotes no occurrence of these risk factors. The total of 9881 patients were enrolled in the study period. Forty-three patients were excluded from the data set due to incomplete registration of the anthropometric outcomes and/or blood sample analyses (n = 27), or subsequent to the diagnosis of type 1 diabetes (n = 16). As a result, 9838 cardiac surgery patients were included in the analyses.

no statistically significant periodicity (p = 0.30, p = 0.58, p = 0.51, and p = 0.75 for aortic stenosis, mitral insufficiency, coronary artery disease, and coronary artery disease with mitral insufficiency, respectively). Conversely, statistically significant seasonal variations for the monthly aggregated data were observed for T2DM (p<0.02), smoking (p<0.001), and elderly (p<0.001) patients alone. Therefore, further analyses were based on these significant variables alone, and their pairwise and combined coexistence were also examined.

## Seasonal variabilities in cardiac surgery patients

Seasonal patterns of statistically significant factors (T2DM alone, smoking alone, and aging alone) for the monthly aggregated data over the 12-year study period are shown in Fig 2. The

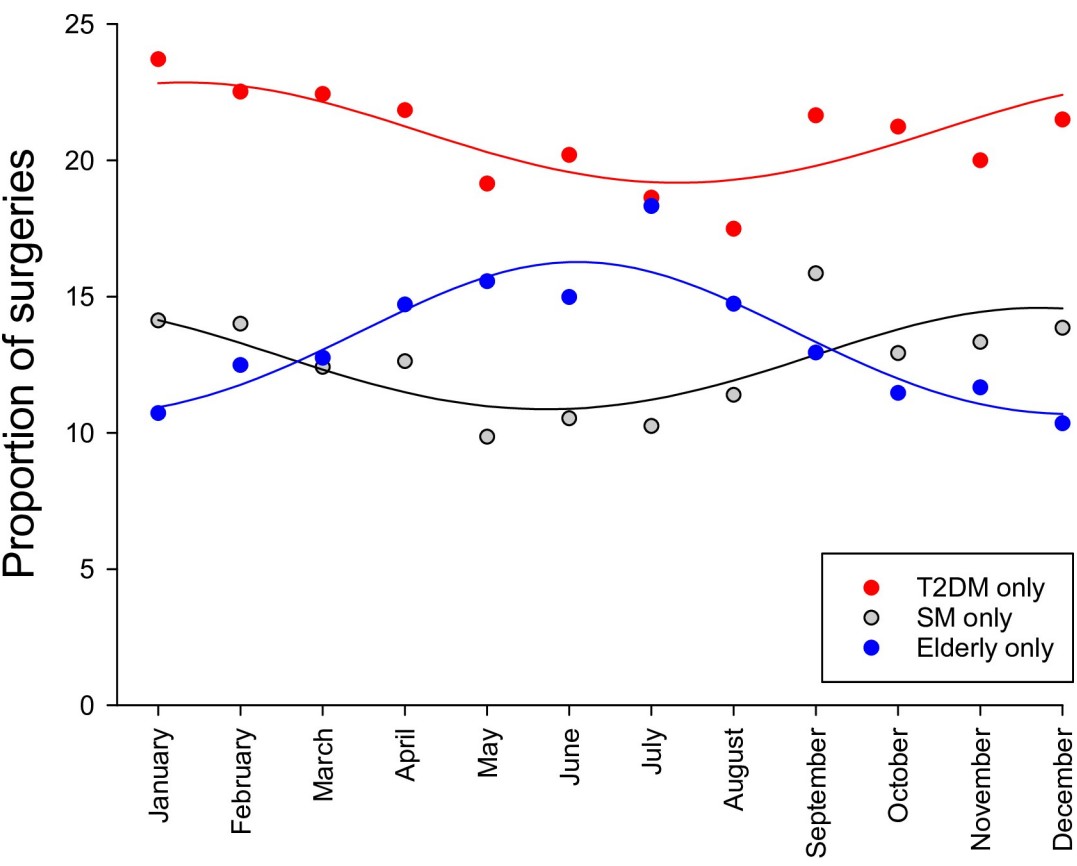

**Fig 2. Seasonal changes in the proportion of surgeries associated with type 2 diabetes mellitus (T2DM only), smoking (SM only), and aging (Elderly only) for the monthly aggregated data over the 12-year study period (January 1, 2007 to December 31, 2018).**

proportion of cardiac surgeries in patients who had T2DM or smoking peaked during the winter months and decreased in the summer. Conversely, the seasonal peak for elderly patients was observed in the summer and was lowest in the winter months.

Seasonal variations in the proportion of cardiac surgeries for patients with paired combinations of the significant factors are shown in Fig 3. No statistically significant seasonal variations were observed in elderly patients with T2DM (p = 0.66) or smoking (p = 0.46). However, the apparent seasonal variations of T2DM and smoking were additive, resulting a marked and statistically significant effect with peak occurrences of these patients in winter and lower in the summer months.

Table 1 summarizes the main parameters of the seasonal variations observed for the statistically significant individual factors (T2DM, smoking, and aging) and their combination (T2DM and smoking) that demonstrated statistically significant seasonality (p<0.001). The goodness of fit of a simple harmonic trend to the data was excellent (>0.9) for the seasonality in T2DM, elderly, and smoking T2DM patients who underwent cardiac surgery. The model fit was worse for seasonal change for smoking only patients; however, a highly significant periodic trend was still observed. The proportion of cardiac surgery patients with T2DM and smoking peaked in January and December, respectively, whereas elderly patients most frequently underwent cardiac surgeries in June. To express the magnitude of seasonal differences in the observed risk factors for cardiac surgery, the amplitudes of each factor relative to the mean rate ([peak—mean]/mean) and to the nadir ([peak—nadir]/nadir) were also calculated.

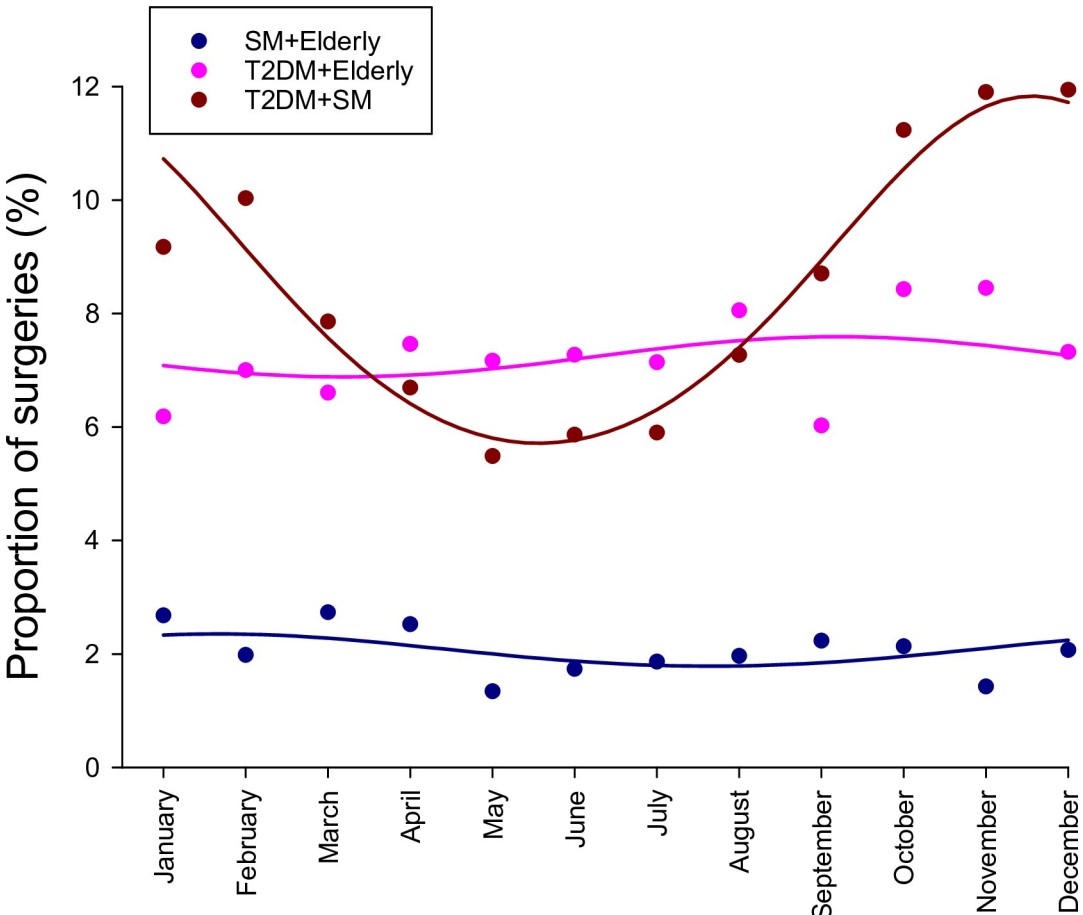

**Fig 3. Seasonal changes in the proportion of surgeries associated with combined smoking and aging (SM + Elderly), type 2 diabetes mellitus and aging (T2DM + Elderly), and type 2 diabetes mellitus and smoking (T2DM + SM) for the monthly aggregated data over the 12-year study period (January 1, 2007 to December 31, 2018).**

The greatest seasonal variability was observed for the relative proportion of smoking patients with T2DM, with values indicating that the rate of such patients at the cardiac surgery unit was more than double in November compared with that in May. The magnitude of seasonal variations in the proportion of cardiac surgeries associated with elderly, smoking, and T2DM

**Table 1. Characteristic parameters of seasonality for statistically significant variables.**

|  | T2DM alone | SM alone | Elderly alone | SM + T2DM |
|---|---|---|---|---|
| **Significance** | p = 0.0184 | p<0.001 | p<0.001 | p<0.001 |
| **Goodness of fit** | 0.92 | 0.51 | 0.95 | 0.97 |
| **Peak (month)** | January | December | June | November |
| **Nadir (month)** | July | May | December | May |
| **(peak–mean)/mean (%)** | 9.6 | 16.4 | 20.3 | 42.6 |
| **(peak–nadir)/nadir (%)** | 19.2 | 34.3 | 52.1 | 107.0 |
| **Maximum increase in proportion of cardiac surgeries (month)** | October | August | March | September |

Maximum increase in the proportion of cardiac surgeries associated with the different pathologies refers to the peak of the first derivative of the fitted seasonality curves. T2DM: type 2 diabetes mellitus; SM: smoking.

patients were lower, but still demonstrated a markedly increased relative risk for cardiac surgeries in the corresponding peak periods.

## Subgroup analyses: Risk factors for cardiovascular complications

S1 Fig in S1 File demonstrates the seasonal variations in the potential risk factors for cardiovascular complications, such as the systolic and diastolic arterial blood pressure, and serum levels of triglyceride, total cholesterol and glucose in group of patients with significant factors for seasonal changes (T2DM alone, smoking alone, and aging alone). Averaging data over the 12-year study period revealed significant seasonal changes in systolic (*[peak-nadir]/nadir*: 6.1%, p<0.001, with peak in February-March) and diastolic blood pressures (4.4%, p<0.05, with peak in January) and serum triglyceride in diabetic patients (17.1%, p<0.005, with peak in December). Significant seasonal variations were also observed in systolic and diastolic blood pressures in smoking patients (6.1% and 6.5%, respectively, p<0.001 for both, with peaks in February), and serum cholesterol in elderly patients (9.1%, p<0.001, with peak in February).

## Discussion

The present study analyzed the medical records of all adult consecutive patients in the past 12-year period at the cardiac surgery unit in our tertiary-care university hospital. Our analyses revealed that the monthly proportion of patients undergoing cardiac surgery with diabetes, smoking, and elderly age exhibited seasonal variation. Non-elderly patients with diabetes and/or smoking showed a peak proportion rate during the winter, whereas heart surgery in elderly patients without diabetes and smoking was most frequently required in the summer. Concomitant occurrence of diabetes and smoking had an additive effect on the proportion of cardiac surgeries associated with the observed pathologies, while the simultaneous presence of older age and diabetes or smoking eliminated the seasonal variation.

Emphases were made on the accuracy and adequacy of data registration. The risk factors examined for seasonal changes (diabetes, smoking and elderly) were identified based on objective, and internationally well-defined diagnostic criteria. Data registration was performed by a stable staff of clinicians including five specialized anesthesiologists during the whole study period on the day prior to the surgery. Since the standard practice at our institution is to avoid waiting lists longer than five days, the seasonal trends observed in the present study accurately reflect the worsening and exacerbation of cardiovascular diseases requiring surgical interventions.

One of the main findings of the present study was a significant elevation in the proportion of cardiac surgeries associated with diabetes during the coldest months of the year. The sinusoidal seasonal trend suggested that the relative risk for patients with diabetes undergoing cardiac surgery during the winter period is almost 20% higher than that in the summer (Fig 2 and Table 1). In patients with diabetes, hyperglycemia leads to endothelial dysfunction, resulting in low-grade inflammatory, prothrombotic, proliferative, and vasoconstrictive processes [22]. These mechanisms may converge and lead to hypertension, atherosclerotic cardiovascular disease, and heart failure [2, 3, 23]. Hypertension may be worsened in a cold environment [4, 24, 25], elevating the myocardial workload and myocardial oxygen demand, or exacerbating functional valve insufficiencies. In addition to these mechanisms, viral infections [26] and/or vitamin deficiency [27] may also be involved. This seasonality is reflected in the high seasonal variation in serum glucose level [28] and the incidence rate of type 1 [26] and T2DM [29, 30] during the coldest months. The systolic and diastolic blood pressures at admission along with the serum triglyceride was significantly higher the present T2DM cohort in January than July over the 12-year study period (S1 Fig in S1 File). Severe manifestation of all these pathologies

requires surgical intervention more frequently during winter for coronary and aortic valve diseases (S1 Table in S1 File).

The proportion of smoking patients undergoing cardiac surgeries without T2DM or old age was lower than those with T2DM alone in our population. Since the seasonal changes were similar in smoking patients and patients with T2DM only, the relative peak-to-peak seasonal variability reached 34% (Fig 1 and Table 1). Similar to T2DM, smoking is also characterized by a blunted response to endothelium-dependent vasodilators due to a diminished bioavailability of nitric oxide [31]. Therefore, the mechanisms also triggered by endothelial dysfunction and subsequent elevated vascular tone may be responsible for the seasonal variations of smoking patients in the cardiac surgery population. Common pathophysiological processes in T2DM and smoking responsible for the seasonality were confirmed by the additive effect of these factors, as it is also reflected in the systolic and diastolic blood pressures in the coldest season (S1 Fig in S1 File). Therefore, the relative peak-to-peak seasonal variability reached more than 100% in smoking patients with diabetes (Fig 3 and Table 1).

A further significant seasonal variability in the cardiac surgery cohort was observed for elderly patients without T2DM or smoking with peak-to-peak seasonal variability >50% (Fig 1 and Table 1). In contrast with smoking and T2DM patients, the proportion of elderly patients undergoing cardiac surgeries peaked in summer. This opposite trend in morbidity may be attributed to compromised elasticity of large conductive arteries [32]. Stiffening of the large arteries makes elderly people susceptible to hypovolemia and hypotension [33]. While seasonal variations were masked by the multifactorial comorbidities of this elderly cohort, the significantly lower diastolic blood pressure observed in the summer months (73.1 mmHg in June-August) compared with winter season (76.5 mmHg in December-February, p<0.05) is in agreement with previous findings, demonstrating that exacerbation of symptoms is expected to be more frequent during the warmest season.

Interestingly, seasonal variability disappeared if aging was associated with diabetes or smoking (Fig 3). The lack of seasonality in these comorbidities may be attributed to the superposition of two sinusoidal waves of aging and diabetes or smoking. Since these waves have similar a period but opposite phases, the periodicity is eliminated. While the lack of season-dependent periodicity in these patients mimics an invariable monthly proportion, these patients are still exposed to both individual risk factors of aging, diabetes, or smoking.

There was no evidence for seasonal changes in the proportion of patients undergoing surgeries for their specific heart diseases (i.e., aortic stenosis, mitral insufficiency, or coronary artery disease) for the whole population. Seasonal variation may be related to peripheral vasculature sensitivity to temperature changes rather than the type of cardiac disease. This suggests that diabetes, smoking, and aging are the primary season-dependent factors regardless of the nature of heart pathology. Gender and BMI do not directly affect the peripheral vasculature, as these factors exhibited no seasonal appearance.

An important feature of our findings is related to local climate. Hungary is situated in East-Central Europe and has four seasons with a continental climate. The Hungarian Meteorological Service calculated the average monthly temperature from daily averages, which varied between 0.7˚C (33.3˚F) in January and 23.1˚C (73.6˚F) in July during the 12-year study period in our region (Fig 4). Our findings may represent the seasonal changes of cardiovascular comorbidities of diabetes, smoking, and aging in the temperate climate zone of the world where the majority of the human population resides.

Some limitation related to the present findings warrant consideration. Hungarian health insurance system provides benefit coverage to all citizens. In addition, free medication is available for low-income patients. These factors blunt the influence of different financial background on patient care. However, the continuous health control may be looser in some

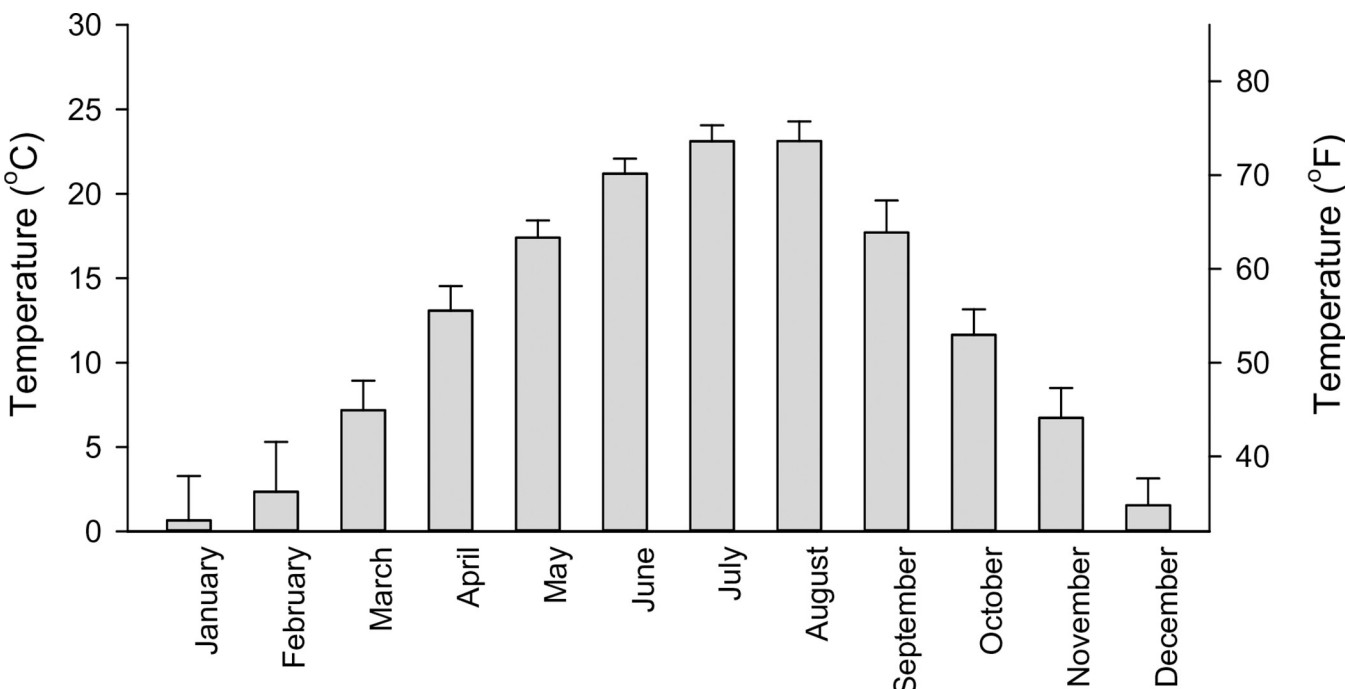

**Fig 4. Monthly temperature (mean and SD) calculated from the daily averages according to the Hungarian Meteorological Service for the 12-year study period (January 1, 2007 to December 31, 2018) in South-East Hungary.**

patients with social negligence that may be a biasing factor in the cardiovascular effects of diabetes. This biasing effect is expected to play a minor role in our findings due to involvement of large cohort for an extended period of time. However, generalization of our findings to other regions with other social and health care systems requires the consideration of local socioeconomic factors, national and institutional scheduling policies.

## Conclusions

In conclusion, analyses of the monthly proportion of cardiac surgeries associated with diabetes, smoking, and elderly age showed seasonal variations, demonstrating a periodicity in exacerbation of cardiovascular diseases requiring surgical intervention. Matching the intensity and control rate of care to these periodicities could prevent worsening of cardiovascular status in diabetes, elderly, and smoking patients. Cardiovascular risk factors should be systematically assessed at least twice per year in diabetes, smoking, and elderly patients. Cardiovascular assessments should primarily occur during the fall–winter period in diabetes and smoking patients, and the spring–summer season in elderly people. Optimal management of patients with diabetes require more frequent cardiovascular risk assessment than the recommended annual control [2, 15], at least three times a year with a focus on the blood pressure and triglyceride assessments. Considering the seasonal trends for risk factors affecting at least half whole population may improve patient and social health care.

## Supporting information

**S1 Checklist. CONSORT 2010 checklist of information to include when reporting a randomised trial.**
(DOC)

**S1 Data. Data-repository.**
(XLSX)

**S1 File.**
(DOCX)

## Author Contributions

**Conceptualization:** Ferenc Peták, Roberta Südy, Csaba Lengyel, Barna Babik.

**Data curation:** Ferenc Peták, Barbara N. Kovács, Szilvia Agócs, Katalin Virág, Tibor Nyári, Andrea Molnár, Roberta Südy, Barna Babik.

**Formal analysis:** Ferenc Peták, Barbara N. Kovács, Katalin Virág, Tibor Nyári, Andrea Molnár, Roberta Südy, Barna Babik.

**Funding acquisition:** Ferenc Peták, Barna Babik.

**Investigation:** Ferenc Peták, Szilvia Agócs, Andrea Molnár, Roberta Südy, Csaba Lengyel, Barna Babik.

**Methodology:** Ferenc Peták, Barbara N. Kovács, Szilvia Agócs, Katalin Virág, Tibor Nyári, Andrea Molnár, Roberta Südy, Barna Babik.

**Project administration:** Barna Babik.

**Resources:** Barna Babik.

**Software:** Katalin Virág, Tibor Nyári.

**Supervision:** Ferenc Peták, Szilvia Agócs, Katalin Virág, Tibor Nyári, Roberta Südy, Csaba Lengyel, Barna Babik.

**Validation:** Ferenc Peták, Katalin Virág, Roberta Südy, Csaba Lengyel, Barna Babik.

**Visualization:** Ferenc Peták, Barna Babik.

**Writing – original draft:** Ferenc Peták, Barbara N. Kovács, Csaba Lengyel, Barna Babik.

**Writing – review & editing:** Ferenc Peták, Barbara N. Kovács, Szilvia Agócs, Katalin Virág, Tibor Nyári, Andrea Molnár, Roberta Südy, Csaba Lengyel, Barna Babik.

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
