## [Decision Letter · Decision Letter 0]

26 Apr 2022

PONE-D-21-33749

SEASONAL CHANGES IN INCIDENCE OF PATIENTS WITH DIABETES UNDERGOING CARDIAC SURGERY

PLOS ONE

Dear Dr. Petak,

Thank you for submitting your manuscript to PLOS ONE. After careful consideration, we feel that it has merit but does not fully meet PLOS ONE’s publication criteria as it currently stands. Therefore, we invite you to submit a revised version of the manuscript that addresses the points raised during the review process.

We look forward to receiving your revised manuscript.

Kind regards,

Chengming Fan, MD, PhD

Academic Editor

PLOS ONE

a) Did participants provide their written or verbal informed consent to participate in this study?

“NO”

5. Thank you for stating the following in the Funding Section of your manuscript:

“This research was supported by a Hungarian Basic Research Council Grant (OTKA-NKFIH K138032) and a GINOP-2.3.2-15-2016-00006 grant.”

“NO”

6. Thank you for stating the following in your Competing Interests section: 

“NO”

7. Please note that in order to use the direct billing option the corresponding author must be affiliated with the chosen institute. Please either amend your manuscript to change the affiliation or corresponding author, or email us at plosone@plos.org with a request to remove this option.

Additional Editor Comments:

Please response to the reviewers point by point.

Reviewers' comments:

Reviewer's Responses to Questions

**Comments to the Author**

1. Is the manuscript technically sound, and do the data support the conclusions?

Reviewer #1: Yes

Reviewer #2: No

Reviewer #3: Partly

2. Has the statistical analysis been performed appropriately and rigorously? 

Reviewer #1: Yes

Reviewer #2: I Don't Know

Reviewer #3: No

3. Have the authors made all data underlying the findings in their manuscript fully available?

Reviewer #1: Yes

Reviewer #2: Yes

Reviewer #3: No

4. Is the manuscript presented in an intelligible fashion and written in standard English?

Reviewer #1: Yes

Reviewer #2: Yes

Reviewer #3: Yes

5. Review Comments to the Author

Reviewer #1: The author indicated the seasonal chaneges in the incidence of cardiac surgery in the patients with diabetes, smoking and/or aging. One of the main findings of the study was a significant elevation of the incidence of patients with diabetes during the coldest months of the year. They speculated the mechanism using several references, and indicated the possible concerning about high blood pressure and triglyceride.

The authors state that there was no seasonal change due to the difference in heart surgery, but isn't there seasonality in the amount of heart surgery? That point should be stated.

The authors should state what they should be especially careful about when conducting seasonal examinations for DMs, smokers, and the elderly.

Reviewer #2: As stated in the title and elsewhere in the manuscript the goal of the authors was to analyze the seasonal variation in the incidence of patients with diabetes undergoing cardiac surgery.

I have several comments to make about this manuscript but the major one is about the definition the authors provide for incidence rate (methods page 6): “ incidence rates were calculated as the number of patients with a given risk factor … divided by the total number of patients in the same month.” The authors use diabetes, advanced age and smoking as outcomes and consider the entire population of hospitalized patients in a given month as the population exposed, the population at risk of developing such outcomes. This is plainly wrong. A more precise description of the authors’ work is a description of seasonal changes in the distribution of risk factors. Only percentages are provided therefore we do not know if the actual number of patients, those with DM for example, increases or decreases in a given month.

Second, the authors provide only p values both in the abstract and the results. As if p values were the only numerical measure worth reporting. An actual numerical measure should be provided for each one of the results, not just their p values.

Methods are insufficiently described. For example the last sentence in the abstract method section belong to the discussion. The authors should clarify the regression methods they used and the results they are referring to. Please clarify exclusion criteria both in the flow chart and the methods. Please explain the following sentence in Methods: “Since patient records were discarded in case of emergency reoperations as a consequence of tamponade or acute bleeding, patients were included in the analyses only after a primary or redo open heart surgeries”.

“Since there was no difference in cardiovascular status between T2DM patients treated with insulin or oral antidiabetics…” How was this assessed?

Results: “In accordance with the diagnostic criteria, HbA1c was significantly higher in patients with diabetes…” This is not a result. Since higher A1c was used to define patients with DM, by definition it is going to be higher in these patients. Why the group None from the flow chart was not included in the figures. I am unclear about which results were achieved from regression vs. stratification.

Reviewer #3: Thank you for inviting me to review this manuscript. This study aims to determine seasonal trends of aging, diabetes, and smoking in a representative sample size from a single center in Hungary. The results showed higher incidence of non-elderly patients with diabetes and smoking during winter periods, while elderly population was predominant in summer periods. This study is interesting and novel, however there are several methodological flaws that reduces its utility in clinical practice. Although personally I am not convinced that season influences the volume of perioperative comorbidities, this study might elucidate further ideas/research in this area. Below I have made some comments that would improve the quality of the manuscript.

Major comments

(1) Abstract: Primary objective should be explicitly written in the abstract and introduction. Please mention what your primary outcome was? I am confused with this statement “Potential additive or subtractive effects of the coexistence of these factors” please re-write and clarify what effects were measured. Authors should provide more details in methods: what statistical technique did they use to measure seasonal effect? Authors should give more than p values.. what effect size did they use?

(2) Introduction: authors should specify the rationale of this study. What is the primary intention of discovering seasonal changes of commorbidities among cardiac surgery patients?

(3) Statistical analysis: the description is incomplete and inconsistent. Please explain how did you measure “significant seasonal change” and “season variability”, provide the effect size that you used, and what method was used to estimate P values. All this information should be clearly stated in the manuscript. Be aware that P values do not give a complete picture and have several limitations. Additionally, the authors did not explain what technique was used for the charts (figures - modeling).

(4) The authors mentioned “goodness of fit” method in results, but there is no explanation of this in methods.

(5) Several limitations should be discussed more broadly in the discussion. For instance, the lack of adjustment for socioeconomic factors, national policies, and institutional scheduling policies.

(6) The seasonal changes of systolic/diastolic pressures in diabetic patients and smokers is interesting and should be described in more detail. I would encourage the authors to present these results in a separate paragraph “Subgroup Analysis”.

(7) Discussion: “This opposite trend in mortality may be attributed to compromised elasticity” Authors did NOT present any data of mortality in this manuscript. Please be consistent.

6. PLOS authors have the option to publish the peer review history of their article (what does this mean?). If published, this will include your full peer review and any attached files.

Reviewer #1: No

Reviewer #2: No

Reviewer #3: No

---

## [Author Response · Author response to Decision Letter 0]

14 Jun 2022

Reviewer #1:

“The author indicated the seasonal chaneges in the incidence of cardiac surgery in the patients with diabetes, smoking and/or aging. One of the main findings of the study was a significant elevation of the incidence of patients with diabetes during the coldest months of the year. They speculated the mechanism using several references, and indicated the possible concerning about high blood pressure and triglyceride.”

Reply 1: We thank the Reviewer for the thoughtful comments contributing to the clarification of these important methodological details.

“The authors state that there was no seasonal change due to the difference in heart surgery, but isn't there seasonality in the amount of heart surgery? That point should be stated.”

Reply 2: This methodological detail is addressed in detail by examining thoroughly the monthly changes in the amount of heart surgery. As the figure demonstrates, there is no seasonal trend in the data in this respect. Only the holyday seasons (July-August and December) show a decrease in the number of patients due to the scheduling the cardiac surgeries by taking into account the availability of human resources available at the university hospital. Please consider that seasonality was demonstrated in the relative incidence rate, which is independent from the total number of patients and thus, the drops in the holyday seasons have no impact on the conclusions of the paper. It is also important to note that this decrease in the total number of patients was not associated with statistically significant differences (p=0.21 by chi-square test) in the relative number of cardiac surgery patients with different diagnoses (see last columns of the table). This conclusion is further confirmed by the sensitivity analyses asked by Reviewer 2. The revised manuscript includes this point (page 7 bottom, page 8 top) in the main manuscript and Table S2 in the online data supplement). 

“The authors should state what they should be especially careful about when conducting seasonal examinations for DMs, smokers, and the elderly.”

Reply 3: The manuscript has been extended by this suggestion of the Reviewer by stating the critical factors when conducting seasonal examinations. Emphases were made on the accuracy and adequacy of data registration. The risk factors examined for seasonal changes (diabetes, smoking and elderly) were identified based on objective, and internationally well-defined diagnostic criteria. Data registration was performed by a stable staff of clinicians including five specialized anesthesiologists during the whole study period on the day prior to the surgery. A further important factor is the lack of waiting list in our institution, since patient flow was not limited by infrastructural available from the human of other resource side. These considerations has been added to the revised manuscript (page 10, bottom).

Reviewer #2:

“As stated in the title and elsewhere in the manuscript the goal of the authors was to analyze the seasonal variation in the incidence of patients with diabetes undergoing cardiac surgery.”

Reply 1: We thank the Reviewer for the thorough revision of our paper and for the pertinent comments contributing to the improvement of our manuscript. Please find below our replies and clarification in a point-by-point fashion.

“I have several comments to make about this manuscript but the major one is about the definition the authors provide for incidence rate (methods page 6): “ incidence rates were calculated as the number of patients with a given risk factor … divided by the total number of patients in the same month.” The authors use diabetes, advanced age and smoking as outcomes and consider the entire population of hospitalized patients in a given month as the population exposed, the population at risk of developing such outcomes. This is plainly wrong. A more precise description of the authors’ work is a description of seasonal changes in the distribution of risk factors. Only percentages are provided therefore we do not know if the actual number of patients, those with DM for example, increases or decreases in a given month.”

Reply 2: We thank the Reviewer for noting the need for clarification of the population description. It was indeed not clearly stated in the original manuscript that our seasonality analyses were performed only on new patients for cardiac surgeries in each month. Therefore, we have used incidence rather than prevalence in the manuscript, as the latter would have suggested the total number of cases in a given timeframe. This consideration was elaborated by adding more details to the description of the data analyses (page 6). 

In agreement with the Reviewer’s suggestion, we have re-analyzed the results by modifying the population to carry out sensitivity analyses. First, we have constructed a base population in which all patients with “T2DM alone”, “T2DM and smoking” and “smoking alone” were excluded. Then we have added “T2DM alone”, “T2DM and smoking” and “smoking alone” as single factor to the population in the new seasonality model. Secondly, we have constructed another base population in which all patients with “T2DM and smoking”, “T2DM alone”, “smoking alone” and “elderly age” were excluded. The seasonality analyses were then carried out, where we have added to the base population only a single variable among those which were excluded. That is, “smoking alone”, “T2DM and smoking”, “T2DM alone” and “elderly age” variables were also analyzed using this reconstructed population. These models have revealed similar significant peaks for seasonality as compared to those reported in the original manuscript for “smoking alone”, “T2DM and smoking”, “T2DM alone” variables. However, there is a slight shift in peak of seasonality for the “elderly age” variable. Thus, the sensitivity analysis is in agreement with our reported results and confirms our main messages. We attach the statistical output of these additional analyses for the interest of the Reviewer (results of the Walter-Elwood method). These outputs also include the actual number of patients for a given month. 

“Second, the authors provide only p values both in the abstract and the results. As if p values were the only numerical measure worth reporting. An actual numerical measure should be provided for each one of the results, not just their p values.”

Reply 3: Based on the Reviewer’s comment, numerical values have been added to the Abstract. We were aiming at avoiding the presentation of the same data in both a table/figure and in the textual Results section of the manuscript, since this may be generally considered as redundant data report. In agreement with the Reviewer’s comment, the entire Results section was revised to include the most important numerical outcomes, where relevant. Please note that where the statement is related to the presence or absence of a statistical significance, reporting a p value summarizes the findings appropriately.

“Methods are insufficiently described. For example the last sentence in the abstract method section belong to the discussion.“

Reply 4: The quoted sentence describes an important methodological detail relevant to our data analyses, and we sought to clarify this point in the Abstract, not only in the manuscript body. In the presence of a waiting list, the relative incidence of the different disorders (e.g., T2DM, smoking and elderly) in the cardiac surgery population would not reflect exacerbation of cardiovascular disorders and would have biased our results accordingly. Based on the Reviewers comment, this sentence was removed from the Abstract.

“The authors should clarify the regression methods they used and the results they are referring to. “

Reply 5: We thank the Reviewer to point out the need for a more detailed description of the statistical method. This part of the manuscript (page 6) has been expanded substantially to clarify that geometric models were used to investigate seasonality by assuming that seasonal fluctuations of an event occur on a fixed date every year and might be described using cyclic patterns over a period of time. The deviance statistic was used to check a goodness of fit for negative binomial regression models.

“Please clarify exclusion criteria both in the flow chart and the methods.“

Reply 6: The exclusion criteria were clarified both in the main text of the manuscript (page 4 and 7) and in the legend of Fig. 1 (page 17).

“Please explain the following sentence in Methods: “Since patient records were discarded in case of emergency reoperations as a consequence of tamponade or acute bleeding, patients were included in the analyses only after a primary or redo open heart surgeries”.”

Reply 7: The rationale for this methodological statement was indeed not clear in the original manuscript. Taking into account surgical procedures related to tamponade and/or bleeding in the relative incidence of the different disorders (e.g., T2DM, smoking and elderly) in the cardiac surgery population would not reflect exacerbation of cardiovascular disorders, as it is the case for other type of primary or redo open heart surgeries. Accordingly, these cases were not assessed for eligibility to avoid the potential bias of these correction surgeries on the definition of exact incidence. Based on the Reviewers comment, the relevant part of the manuscript has been expanded to clarify this methodological consideration (page 4).

“ “Since there was no difference in cardiovascular status between T2DM patients treated with insulin or oral antidiabetics…” How was this assessed?”

Reply 8: We thank the Reviewer for raising the need for evidence in this statement. Systematic assessment of the cardiovascular status of T2DM patients treated with insulin or oral antidiabetics would have required prospective analyses with a need of large cohort due to the clinician- and patient-related variabilities. Thus, such measurements were not in the focus of the present study. This sentence has been moderated accordingly (page 5, top). 

“Results: “In accordance with the diagnostic criteria, HbA1c was significantly higher in patients with diabetes…” This is not a result. Since higher A1c was used to define patients with DM, by definition it is going to be higher in these patients. Why the group None from the flow chart was not included in the figures. I am unclear about which results were achieved from regression vs. stratification.”

Reply 9: We fully agree with the Reviewer on the questionable mixture of the diagnostic criteria and the Results. We thought that reporting the exact HbA1c is informative in assessing the difference between the patients with or without diabetes, however it makes no sense to perform a statistical analysis on this outcome. Accordingly, the questioned statistical outcome was omitted from the revised Results section (page 7).

As concerns the question of inclusion of data on the charts, please consider that only those variables were presented on the figures where significant seasonal variations for the monthly aggregated data were observed, i.e., T2DM (p<0.02), smoking (p<0.001), and elderly (p<0.001) patients alone and their combined coexistence. This is clarified in the Methods section of the paper (page 8, paragraph 2). Including other groups of patients on the graphs with no seasonal variation (e.g., group “None”) would somewhat overload the figures and thereby divert the focus of the readers from the main message.

 

Reviewer #3:

“Thank you for inviting me to review this manuscript. This study aims to determine seasonal trends of aging, diabetes, and smoking in a representative sample size from a single center in Hungary. The results showed higher incidence of non-elderly patients with diabetes and smoking during winter periods, while elderly population was predominant in summer periods. This study is interesting and novel, however there are several methodological flaws that reduces its utility in clinical practice. Although personally I am not convinced that season influences the volume of perioperative comorbidities, this study might elucidate further ideas/research in this area. Below I have made some comments that would improve the quality of the manuscript.”

Reply: We thank the Reviewer for the thorough revision of the manuscript and for the thoughtful comments contributing greatly to the improvement of the paper.

“Major comments

“(1) Abstract: Primary objective should be explicitly written in the abstract and introduction. Please mention what your primary outcome was? I am confused with this statement “Potential additive or subtractive effects of the coexistence of these factors” please re-write and clarify what effects were measured. Authors should provide more details in methods: what statistical technique did they use to measure seasonal effect? Authors should give more than p values.. what effect size did they use?” “

Reply 1: We thank the Reviewer for raising the need for this important clarification. The primary objective of the present study is expressed more explicitly in the revised version of the Abstract (page 2) and the Background (page 3 bottom and page 4 top), in agreement with the Reviewer’s request. The examination of the additive and subtractive factors was also clarified in the Abstract by rewording this part of the abstract with avoiding these ambiguous terminologies (page 3). Further details are included in the revised manuscript about the statistical technique to measure seasonal effect (page 6). Furthermore, the entire Abstract and Results sections were revised to report our findings in a more detailed manner than giving p values only. In accordance with the Reviewer’s comment, the statistical technique to measure seasonal effects is described in a more detailed manner in the revised manuscript with citing more relevant literature (page 6, paragraph 1), and the effect size is also specified (page 6).

“(2) Introduction: authors should specify the rationale of this study. What is the primary intention of discovering seasonal changes of commorbidities among cardiac surgery patients?”

Reply 2: The Introduction has been rewritten to state clearly the rationale of the study (page 3 bottom and page 4 top).

“(3) Statistical analysis: the description is incomplete and inconsistent. Please explain how did you measure “significant seasonal change” and “season variability”, provide the effect size that you used, and what method was used to estimate P values. All this information should be clearly stated in the manuscript. Be aware that P values do not give a complete picture and have several limitations. Additionally, the authors did not explain what technique was used for the charts (figures - modeling).”

Reply 3: We thank the Reviewer for noting the need for a more detailed description of the statistical analyses. We fully agree that p values do not give a complete picture and have several limitations and thus, we added the key figures to the Abstract and the Results section.

In our study, the geometric models were used to investigate seasonality, which was introduced by Edwards in 1961 [1]. Here, we assume that seasonal fluctuations of an event occur on a fixed date every year and might be described using cyclic patterns over a period of time. Walter and Elwood developed this model by adding the population at risk [2]. Stolwijk et al described the use of this method for general linear models [3]. We have used in our analyzes both Walter-Elwood test and negative binomial regression methods. The power of the Walter-Elwood test is 100% [4-5]. These details have been added to the revised manuscript (page 6). 

As concerns the Reviewer’s comment on the effect size, please consider that if statistical power is high, the likelihood of deciding there is an effect, when one does exist, is high [6]. 

The charts were prepared by using SigmaPlot software package (Version 13, Systat Software, Inc. Chicago, IL, USA); this is now specified in the revised manuscript (page 6).

[1] Edwards, J. H. (1961): The recognition and estimation of cyclic trends. – Ann Hum Genet 25: 83-87.

[2] Walter, S. D., Elwood, J. M. (1975): A test for seasonality of events with a variable population at risk. – Br J PrevSoc Med 29: 18-21.

[3] Stolwijk, A.M., Straatman, H., Zielhuis, G. A. (1999): Studying seasonality by using sine and cosine functions in regression analysis. – J Epidemiol Community Health 53: 235-238.

[4] Walter, S. D. (1977): The power of a test for seasonality. – Br J PrevSoc Med 31: 137-140.

[5] Barnett, A. G., Dobson, A. J. (2010): Analysing Seasonal Health Data. – Springer, ISBN 978-3-642-10748-1.

[6] Gail M. Sullivan, MD, MPH and Richard Feinn, PhD. Using Effect Size—or Why the P Value Is Not Enough. Journal of Graduate Medical Education, September 2012

“(4) The authors mentioned “goodness of fit” method in results, but there is no explanation of this in methods.”

Reply 4: We thank the Reviewer for noting this lack of this information in the Methods. In generally, a chi-square test was applied to check a goodness of fit. The deviance statistic was used to check a goodness of fit for negative binomial regression models. Similarly, Walter and Elwood described a goodness of fit calculation [2] which was also applied. All analyses were carried out using STATA version 17 statistical software to confirm the goodness of fit of models. This methodological detail has been added to the revised manuscript (page 6).

“(5) Several limitations should be discussed more broadly in the discussion. For instance, the lack of adjustment for socioeconomic factors, national policies, and institutional scheduling policies.”

Reply 5: Discussion of the limitation of our findings is an important point. Accordingly, the manuscript has been expanded to include this aspect (page 13 bottom, page 14 top).

“(6) The seasonal changes of systolic/diastolic pressures in diabetic patients and smokers is interesting and should be described in more detail. I would encourage the authors to present these results in a separate paragraph “Subgroup Analysis”.”

Reply 6: We thank for this suggestion, these risk factors for cardiovascular complications are emphasized more in the revised manuscript by including these results in a separate paragraph titled “Subgroup analyses: risk factors for cardiovascular complications” (page 9, bottom).

“(7) Discussion: “This opposite trend in mortality may be attributed to compromised elasticity” Authors did NOT present any data of mortality in this manuscript. Please be consistent.”

Reply 7: We thank for picking up this mistake. We meant to write here “morbidity” instead of mortality, as the latter was indeed not assessed. The sentence has been corrected accordingly (page 12).

---

## [Decision Letter · Decision Letter 1]

5 Jul 2022

PONE-D-21-33749R1SEASONAL CHANGES IN INCIDENCE OF PATIENTS WITH DIABETES UNDERGOING CARDIAC SURGERYPLOS ONE

Dear Dr. Petak,

Thank you for submitting your manuscript to PLOS ONE. After careful consideration, we feel that it has merit but does not fully meet PLOS ONE’s publication criteria as it currently stands. Therefore, we invite you to submit a revised version of the manuscript that addresses the points raised during the review process. Please submit your revised manuscript by Aug 19 2022 11:59PM. If you will need more time than this to complete your revisions, please reply to this message or contact the journal office at plosone@plos.org. Please include the following items when submitting your revised manuscript:A rebuttal letter that responds to each point raised by the academic editor and reviewer(s). You should upload this letter as a separate file labeled 'Response to Reviewers'.A marked-up copy of your manuscript that highlights changes made to the original version. You should upload this as a separate file labeled 'Revised Manuscript with Track Changes'.An unmarked version of your revised paper without tracked changes. You should upload this as a separate file labeled 'Manuscript'.

We look forward to receiving your revised manuscript.

Kind regards,

Chengming Fan, MD, PhD

Academic Editor

PLOS ONE

Additional Editor Comments:

Please response to the reviewers' comments point by point.

Reviewers' comments:

Reviewer's Responses to Questions

**Comments to the Author**

1. If the authors have adequately addressed your comments raised in a previous round of review and you feel that this manuscript is now acceptable for publication, you may indicate that here to bypass the “Comments to the Author” section, enter your conflict of interest statement in the “Confidential to Editor” section, and submit your "Accept" recommendation.

Reviewer #1: (No Response)

Reviewer #2: (No Response)

Reviewer #3: All comments have been addressed

2. Is the manuscript technically sound, and do the data support the conclusions?

Reviewer #1: Yes

Reviewer #2: Partly

Reviewer #3: Yes

3. Has the statistical analysis been performed appropriately and rigorously? 

Reviewer #1: Yes

Reviewer #2: I Don't Know

Reviewer #3: Yes

4. Have the authors made all data underlying the findings in their manuscript fully available?

Reviewer #1: Yes

Reviewer #2: Yes

Reviewer #3: Yes

5. Is the manuscript presented in an intelligible fashion and written in standard English?

Reviewer #1: Yes

Reviewer #2: Yes

Reviewer #3: Yes

6. Review Comments to the Author

Reviewer #1: The author indicated the seasonal changes in the incidence of cardiac surgery in the patients with diabetes, smoking and/or aging. One of the main findings of the study was a significant elevation of the incidence of patients with diabetes during the coldest months of the year. They speculated the mechanism using several references, and indicated the possible concerning about high blood pressure and triglyceride.

The authors are well responding to the reviewer's questions and comments. The manuscript is well revised and suitable for the publication.

Reviewer #2: The authors have greatly improved the manuscript. However an important point in my opinion was not addressed. I appreciate the fact that only new surgeries were taken into account was highlighted and the sensitivity analysis using different base populations, with and w/o the risk factors, but I believe that what the authors describe is not incidence. The denominator for the incidence is the population exposed to the risk of an event. If the event is cardiac surgery and the exposure is type 2 diabetes or smoking, then the population exposed would be all people with diabetes or all smokers. Certainly not all people undergoing cardiac surgery. If the authors consider as the event “a person having diabetes AND undergoing cardiac surgery” then the population at risk would be all people with cardiac disease, with and w/o diabetes.

Even if we make several reasonable assumptions: the need for cardiac surgeries among people without diabetes remains constant; the number of people at risk, with DM or smokers remain constant throughout the change of season, it is unclear why the authors present their results as incidence variations.

For example, if among 100 people undergoing cardiac surgeries in one month there are 30 with diabetes, and the following month the number of people with diabetes who have surgery doubles because of temperature variation, then we would have a proportion of surgeries associated with diabetes, given the above assumptions, from 30% to 46% which is not the measure of incidence variation.

Reviewer #3: Thank you for addressing most of the reviewers' comments. However, the explanation about the effect size still remains unclear. I understand that the percentage of change (variation) is the main effect size for the association between seasonal variation and health parameters. Is that right? Please be more concise and explicit in this regard.

7. PLOS authors have the option to publish the peer review history of their article (what does this mean?). If published, this will include your full peer review and any attached files.

Reviewer #1: No

Reviewer #2: No

Reviewer #3: No

---

## [Author Response · Author response to Decision Letter 1]

13 Jul 2022

Reviewer #2: 

“The authors have greatly improved the manuscript. However an important point in my opinion was not addressed. I appreciate the fact that only new surgeries were taken into account was highlighted and the sensitivity analysis using different base populations, with and w/o the risk factors, but I believe that what the authors describe is not incidence. The denominator for the incidence is the population exposed to the risk of an event. If the event is cardiac surgery and the exposure is type 2 diabetes or smoking, then the population exposed would be all people with diabetes or all smokers. Certainly not all people undergoing cardiac surgery. If the authors consider as the event “a person having diabetes AND undergoing cardiac surgery” then the population at risk would be all people with cardiac disease, with and w/o diabetes.

Even if we make several reasonable assumptions: the need for cardiac surgeries among people without diabetes remains constant; the number of people at risk, with DM or smokers remain constant throughout the change of season, it is unclear why the authors present their results as incidence variations.

For example, if among 100 people undergoing cardiac surgeries in one month there are 30 with diabetes, and the following month the number of people with diabetes who have surgery doubles because of temperature variation, then we would have a proportion of surgeries associated with diabetes, given the above assumptions, from 30% to 46% which is not the measure of incidence variation.”

Reply: We thank the Reviewer for appreciating our efforts to revise our paper based on the previous highly pertinent comments. We are also indebted for highlighting the remaining concern with the use of “incidence” as a potentially misleading terminology. We fully accept the detailed explanation of the correct interpretation of our data and changed the term “incidence” into “proportion of surgeries associated with diabetes/smoking/elderly” throughout in the revised manuscript.

Reviewer #3: 

“Thank you for addressing most of the reviewers' comments. However, the explanation about the effect size still remains unclear. I understand that the percentage of change (variation) is the main effect size for the association between seasonal variation and health parameters. Is that right? Please be more concise and explicit in this regard.”

Reply: We are grateful to the Reviewer for acknowledging our revision and we thank for noting for the need of clarifying this methodological detail. In agreement with the Reviewer’s statement, the percentage of change (variation) is the main effect size for the association between seasonal variation and health parameters. This is now specified explicitly in the revised manuscript (page 6, paragraph 1).

---

## [Decision Letter · Decision Letter 2]

23 Aug 2022

SEASONAL CHANGES IN PROPORTION OF CARDIAC SURGERIES ASOCIATED WITH DIABETES, SMOKING AND ELDERLY AGE

PONE-D-21-33749R2

Dear Dr. Petak,

We’re pleased to inform you that your manuscript has been judged scientifically suitable for publication and will be formally accepted for publication once it meets all outstanding technical requirements.

Kind regards,

Chengming Fan, MD, PhD

Academic Editor

PLOS ONE

Reviewers' comments:

Reviewer's Responses to Questions

**Comments to the Author**

Reviewer #2: All comments have been addressed

Reviewer #3: All comments have been addressed

2. Is the manuscript technically sound, and do the data support the conclusions?

Reviewer #2: Yes

Reviewer #3: Yes

3. Has the statistical analysis been performed appropriately and rigorously? 

Reviewer #2: I Don't Know

Reviewer #3: Yes

4. Have the authors made all data underlying the findings in their manuscript fully available?

Reviewer #2: Yes

Reviewer #3: Yes

5. Is the manuscript presented in an intelligible fashion and written in standard English?

Reviewer #2: Yes

Reviewer #3: Yes

6. Review Comments to the Author

Reviewer #2: All my comments have been addressed satisfactorily by the authors. I have no further recommendations.

Reviewer #3: Thank you for addressing all my comments. This study highlights the importance of seasonal changes of comorbidities. Congratulation on a nice paper.

7. PLOS authors have the option to publish the peer review history of their article (what does this mean?). If published, this will include your full peer review and any attached files.

Reviewer #2: No

Reviewer #3: No

---

## [Editor Report · Acceptance letter]

26 Aug 2022

PONE-D-21-33749R2 

SEASONAL CHANGES IN PROPORTION OF CARDIAC SURGERIES ASOCIATED WITH DIABETES, SMOKING AND ELDERLY AGE 

Dear Dr. Petak:

I'm pleased to inform you that your manuscript has been deemed suitable for publication in PLOS ONE. Congratulations! Your manuscript is now with our production department. 

Kind regards, 

on behalf of

Dr. Chengming Fan 

Academic Editor

PLOS ONE